# FEARNET: BRAIN-INSPIRED MODEL FOR INCREMENTAL LEARNING

**Ronald Kemker and Christopher Kanan**[*]
Carlson Center for Imaging Science
Rochester Institute of Technology
Rochester, NY 14623, USA
{rmk6217,kanan}@rit.edu

## ABSTRACT

Incremental class learning involves sequentially learning classes in bursts of examples from the same class. This violates the assumptions that underlie methods for training standard deep neural networks, and will cause them to suffer from catastrophic forgetting. Arguably, the best method for incremental class learning is iCaRL, but it requires storing training examples for each class, making it challenging to scale. Here, we propose FearNet for incremental class learning. FearNet is a generative model that does not store previous examples, making it memory efficient. FearNet uses a brain-inspired dual-memory system in which new memories are consolidated from a network for recent memories inspired by the mammalian hippocampal complex to a network for long-term storage inspired by medial prefrontal cortex. Memory consolidation is inspired by mechanisms that occur during sleep. FearNet also uses a module inspired by the basolateral amygdala for determining which memory system to use for recall. FearNet achieves state-of-the-art performance at incremental class learning on image (CIFAR-100, CUB-200) and audio classification (AudioSet) benchmarks.

## 1 INTRODUCTION

In incremental classification, an agent must sequentially learn to classify training examples, without necessarily having the ability to re-study previously seen examples. While deep neural networks (DNNs) have revolutionized machine perception (Krizhevsky et al., 2012), off-the-shelf DNNs cannot incrementally learn classes due to catastrophic forgetting. Catastrophic forgetting is a phenomenon in which a DNN completely fails to learn new data without forgetting much of its previously learned knowledge (McCloskey & Cohen, 1989). While methods have been developed to try and mitigate catastrophic forgetting, as shown in Kemker et al. (2018), these methods are not sufficient and perform poorly on larger datasets. In this paper, we propose FearNet, a brain-inspired system for incrementally learning categories that significantly outperforms previous methods.

The standard way for dealing with catastrophic forgetting in DNNs is to avoid it altogether by mixing new training examples with old ones and completely re-training the model offline. For large datasets, this may require weeks of time, and it is not a scalable solution. An ideal incremental learning system would be able to assimilate new information without the need to store the entire training dataset. A major application for incremental learning includes real-time operation on-board embedded platforms that have limited computing power, storage, and memory, e.g., smart toys, smartphone applications, and robots. For example, a toy robot may need to learn to recognize objects within its local environment and of interest to its owner. Using cloud computing to overcome these resource limitations may pose privacy risks and may not be scalable to a large number of embedded devices. A better solution is on-device incremental learning, which requires the model to use less storage and computational power.

In this paper, we propose an incremental learning framework called FearNet (see Fig. 1). FearNet has three brain-inspired sub-systems: 1) a recent memory system for quick recall, 2) a memory

---

[*]Corresponding author.

system for long-term storage, and 3) a sub-system that determines which memory system to use for a particular example. FearNet mitigates catastrophic forgetting by consolidating recent memories into long-term storage using pseudorehearsal (Robins, 1995). Pseudorehearsal allows the network to revisit previous memories during incremental training without the need to store previous training examples, which is more memory efficient.

**Problem Formulation:** Here, incremental class learning consists of $T$ study-sessions. At time $t$, the learner receives a batch of data $B_t$, which contains $N_t$ labeled training samples, i.e., $B_t = \{(\mathbf{x}_j, y_j)\}_{j=1}^{N_t}$, where $\mathbf{x}_j \in \mathbb{R}^d$ is the input feature vector to be classified and $y_j$ is its corresponding label. The number of training samples $N_t$ may vary between sessions, and the data inside a study-session is not assumed to be independent and identically distributed (iid). During a study session, the learner only has access to its current batch, but it may use its own memory to store information from prior study sessions. We refer to the first session as the model's "base-knowledge," which contains exemplars from $M \geq 1$ classes. The batches learned in all subsequent sessions contain only one class, i.e., all $y_j$ will be identical within those sessions.

**Novel Contributions:** Our contributions include:

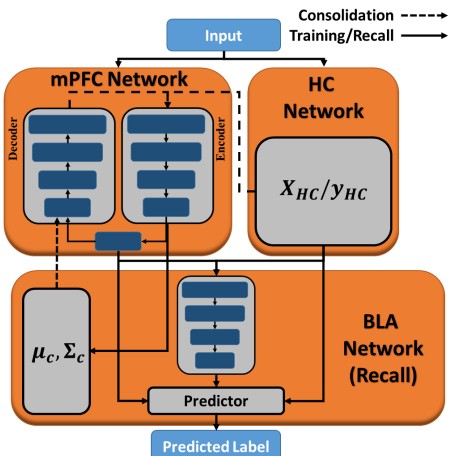

Figure 1: FearNet consists of three brain-inspired modules based on 1) mPFC (long-term storage), 2) HC (recent storage), and 3) BLA for determining whether to use mPFC or HC for recall.

1. FearNet's architecture includes three neural networks: one inspired by the hippocampal complex (HC) for recent memories, one inspired by the medial prefrontal cortex (mPFC) for long-term storage, and one inspired by the basolateral amygdala (BLA) that determines whether to use HC or mPFC for recall.

2. Motivated by memory replay during sleep, FearNet employs a generative autoencoder for pseudorehearsal, which mitigates catastrophic forgetting by generating previously learned examples that are replayed alongside novel information during consolidation. This process does not involve storing previous training data.

3. FearNet achieves state-of-the-art results on large image and audio datasets with a relatively small memory footprint, demonstrating how dual-memory models can be scaled.

## 2  RELATED WORK

Catastrophic forgetting in DNNs occurs due to the plasticity-stability dilemma (Abraham & Robins, 2005). If the network is too plastic, older memories will quickly be overwritten; however, if the network is too stable, it is unable to learn new data. This problem was recognized almost 30 years ago (McCloskey & Cohen, 1989). In French (1999), methods developed in the 1980s and 1990s are extensively discussed, and French argued that mitigating catastrophic forgetting would require having two separate memory centers: one for the long-term storage of older memories and another to quickly process new information as it comes in. He also theorized that this type of dual-memory system would be capable of consolidating memories from the fast learning memory center to long-term storage.

Catastrophic forgetting often occurs when a system is trained on non-iid data. One strategy for reducing this phenomenon is to mix old examples with new examples, which simulates iid conditions. For example, if the system learns ten classes in a study session and then needs to learn 10 new classes in a later study session, one solution could be to mix examples from the first study session into the later study session. This method is known as rehearsal, and it is one of the earliest methods for reducing catastrophic forgetting (Hetherington & Seidenberg, 1989). Rehearsal essentially uses an external memory to strengthen the model's representations for examples learned previously, so

that they are not overwritten when learning data from new classes. Rehearsal reduces forgetting, but performance is still worse than offline models. Moreover, rehearsal requires storing all of the training data. Robins (1995) argued that storing of training examples was inefficient and of "little interest," so he introduced pseudorehearsal. Rather than replaying past training data, in pseudorehearsal, the algorithm *generates* new examples for a given class. In Robins (1995), this was done by creating random input vectors, having the network assign them a label, and then mixing them into the new training data. This idea was revived in Draelos et al. (2017), where a generative autoencoder was used to create pseudo-examples for unsupervised incremental learning. This method inspired FearNet's approach to memory consolidation. Pseudorehearsal is related to memory replay that occurs in mammalian brains, which involves reactivation of recently encoded memories in HC so that they can be integrated into long-term storage in mPFC (Rasch & Born, 2013).

Recently there has been renewed interest in solving catastrophic forgetting in supervised learning. Many new methods are designed to mitigate catastrophic forgetting when each study session contains a permuted version of the entire training dataset (see Goodfellow et al. (2013)). Unlike incremental class learning, all labels are contained in each study session. PathNet uses an evolutionary algorithm to find the optimal path through a large DNN, and then freezes the weights along that path (Fernando et al., 2017). It assumes all classes are seen in each study session, and it is not capable of incremental class learning. Elastic Weight Consolidation (EWC) employs a regularization scheme that redirects plasticity to the weights that are least important to previously learned study sessions (Kirkpatrick et al., 2017). After EWC learns a study session, it uses the training data to build a Fisher matrix that determines the importance of each feature to the classification task it just learned. EWC was shown to work poorly at incremental class learning in Kemker et al. (2018).

The Fixed Expansion Layer (FEL) model mitigates catastrophic forgetting by using sparse updates (Coop et al., 2013). FEL uses two hidden layers, where the second hidden layer (i.e., the FEL layer) has connectivity constraints. The FEL layer is much larger than the first hidden layer, is sparsely populated with excitatory and inhibitory weights, and is not updated during training. This limits learning of dense shared representations, which reduces the risk of learning interfering with old memories. FEL requires a large number of units to work well (Kemker et al., 2018).

Gepperth & Karaoguz (2016) introduced a new approach for incremental learning, which we call Gepp-Net. GeppNet uses a self-organizing map (SOM) to reorganize the input onto a two-dimensional lattice. This serves as a long-term memory, which is fed into a simple linear layer for classification. After the SOM is initialized, it can only be updated if the input is sufficiently novel. This prevents the model from forgetting older data too quickly. GeppNet also uses rehearsal using all previous training data. A variant of Gepp-Net, GeppNet+STM, uses a fixed-size memory buffer to store novel examples. When this buffer is full, it replaces the oldest example. During pre-defined intervals, the buffer is used to train the model. Gepp-Net+STM is better at retaining base-knowledge since it only trains during its consolidation phase, but the STM-free version learns new data better because it updates the model on every novel labeled input.

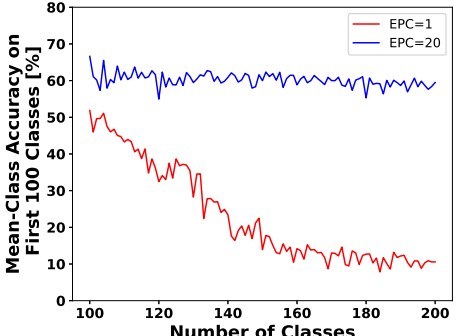

Figure 2: iCaRL's performance depends heavily on the number of exemplars per class (EPC) that it stores. Reducing EPC from 20 (blue) to 1 (red) severely impairs its ability to recall older information.

iCaRL (Rebuffi et al., 2017) is an incremental class learning framework. Rather than directly using a DNN for classification, iCaRL uses it for supervised representation learning. During a study session, iCaRL updates a DNN using the study session's data and a set of $J$ stored examples from earlier sessions ($J = 2,000$ for CIFAR-100 in their paper), which is a kind of rehearsal. After a study session, the $J$ examples retained are carefully chosen using herding. After learning the entire dataset, iCaRL has retained $J/T$ exemplars per class (e.g., $J/T = 20$ for CIFAR-100). The DNN in iCaRL is then used to compute an embedding for each stored example, and then the mean embedding for each class seen is computed. To classify a new instance, the DNN is used to compute an embedding for it, and then the class with the nearest mean embedding is assigned. iCaRL's performance is heavily influenced by the number of examples it stores, as shown in Fig. 2.

## 3    MAMMALIAN MEMORY: NEUROSCIENCE AND MODELS

FearNet is heavily inspired by the dual-memory model of mammalian memory (McClelland et al., 1995), which has considerable experimental support from neuroscience (Frankland et al., 2004; Takashima et al., 2006; Kitamura et al., 2017; Bontempi et al., 1999; Taupin & Gage, 2002; Gais et al., 2007). This theory proposes that HC and mPFC operate as complementary memory systems, where HC is responsible for recalling recent memories and mPFC is responsible for recalling remote (mature) memories. GeppNet is the most recent DNN to be based on this theory, but it was also independently explored in the 1990s in French (1997) and Ans & Rousset (1997). In this section, we review some of the evidence for the dual-memory model.

One of the major reasons why HC is thought to be responsible for recent memories is that if HC is bilaterally destroyed, then anterograde amnesia occurs with old memories for semantic information preserved. One mechanism HC may use to facilitate creating new memories is adult neurogenesis. This occurs in HC's dentate gyrus (Altman, 1963; Eriksson et al., 1998). The new neurons have higher initial plasticity, but it reduces as time progresses (Deng et al., 2010).

In contrast, mPFC is responsible for the recall of remote (long-term) memories (Bontempi et al., 1999). Taupin & Gage (2002) and Gais et al. (2007) showed that mPFC plays a strong role in memory consolidation during REM sleep. McClelland et al. (1995) and Euston et al. (2012) theorized that, during sleep, HC reactivates recent memories to prevent forgetting which causes these recent memories to replay in mPFC as well, with dreams possibly being caused by this process. After memories are transferred from HC to mPFC, evidence suggests that corresponding memory in HC is erased (Poe, 2017).

Recently, Kitamura et al. (2017) performed contextual fear conditioning (CFC) experiments in mice to trace the formation and consolidation of recent memories to long-term storage. CFC experiments involve shocking mice while subjecting them to various visual stimuli (i.e., colored lights). They found that BLA, which is responsible for regulating the brain's fear response, would shift where it retrieved the corresponding memory from (HC or mPFC) as that memory was consolidated over time. FearNet follows the memory consolidation theory proposed by Kitamura et al. (2017).

## 4    THE FEARNET MODEL

FearNet has two complementary memory centers, 1) a short-term memory system that immediately learns new information for recent recall (HC) and 2) a DNN for the storage of remote memories (mPFC). FearNet also has a separate BLA network that determines which memory center contains the associated memory required for prediction. During sleep phases, FearNet uses a generative model to consolidate data from HC to mPFC through pseudorehearsal. Pseudocode for FearNet is provided in the supplemental material. Because the focus of our work is not representation learning, we use pre-trained ResNet embeddings to obtain features that are fed to FearNet.

### 4.1    DUAL-MEMORY STORAGE

FearNet's HC model is a variant of a probabilistic neural network (Specht, 1990). HC computes class conditional probabilities using stored training examples. Formally, HC estimates the probability that an input feature vector $\mathbf{x}$ belongs to class $k$ as

$$P_{HC}\left(C = k|\mathbf{x}\right) = \frac{\beta_k}{\sum_{k'} \beta_{k'}} \tag{1}$$

$$\beta_k = \begin{cases} \left(\epsilon + \min_j \|\mathbf{x} - \mathbf{u}_{k,j}\|_2\right)^{-1} & \text{if HC contains instances of class } k \\ 0 & \text{otherwise} \end{cases} \tag{2}$$

where $\epsilon > 0$ is a regularization parameter and $\mathbf{u}_{k,j}$ is the $j$'th stored exemplar in HC for class $k$. All exemplars are removed from HC after they are consolidated into mPFC.

FearNet's mPFC is implemented using a DNN trained both to reconstruct its input using a symmetric encoder-decoder (autoencoder) and to compute $P_{mPFC}\left(C = k|\mathbf{x}\right)$. The autoencoder enables us to

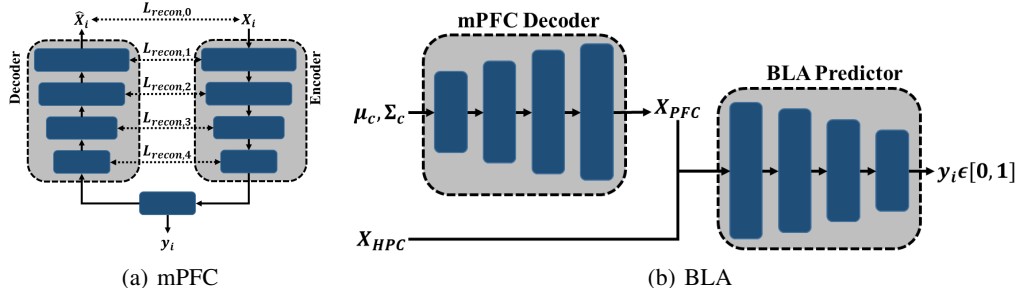

(a) mPFC                                          (b) BLA

Figure 3: The mPFC and BLA sub-systems in FearNet. mPFC is responsible for the long-term storage of remote memories. BLA is used during prediction time to determine if the memory should be recalled from short- or long-term memory.

use pseudorehearsal, which is described in more detail in Sec. 4.2. The loss function for mPFC is

$$\mathcal{L}_{mPFC} = \mathcal{L}_{class} + \mathcal{L}_{recon}, \tag{3}$$

where $\mathcal{L}_{class}$ is the supervised classification loss and $\mathcal{L}_{recon}$ is the unsupervised reconstruction loss, as illustrated in Fig. 3(a). For $\mathcal{L}_{class}$, we use standard softmax loss. $\mathcal{L}_{recon}$ is the weighted sum of mean squared error (MSE) reconstruction losses from each layer, which is given by

$$\mathcal{L}_{recon} = \sum_{j=0}^{M} \sum_{i=0}^{H_j-1} \left\| h_{encoder,(i,j)} - h_{decoder,(i,j)} \right\|_2^2 \tag{4}$$

where $M$ is the number of mPFC layers, $H_j$ is the number of hidden units in layer $j$, $h_{encoder,(i,j)}$ and $h_{decoder,(i,j)}$ are the outputs of the encoder/decoder at layer $j$ respectively, and $\lambda_j$ is the reconstruction weight for that layer. mPFC is similar to a Ladder Network (Rasmus et al., 2015), which combines classification and reconstruction to improve regularization, especially during low-shot learning. The $\lambda_j$ hyperparameters were found empirically, with $\lambda_0$ being largest and decreasing for deeper layers (see supplementary material). This prioritizes the reconstruction task, which makes the generated pseudo-examples more realistic. When training is completed during a study session, all of the data in HC is pushed through the encoder to extract a dense feature representation of the original data, and then we compute a mean feature vector $\mu_c$ and covariance matrix $\Sigma_c$ for each class $c$. These are stored and used to generate pseudo-examples during consolidation (see Sec. 4.2). We study FearNet's performance as a function of how much data is stored in HC in Sec. 6.2.

## 4.2 PSEUDOREHEARSAL FOR MEMORY CONSOLIDATION

During FearNet's sleep phase, the original inputs stored in HC are transferred to mPFC using pseudo-examples created by an autoencoder. This process is known as intrinsic replay, and it was used by Draelos et al. (2017) for unsupervised learning.

Using the class statistics from the encoder, pseudo-examples for class $c$ are generated by sampling a Gaussian with mean $\mu_c$ and covariance matrix $\Sigma_c$ to obtain $\hat{\mathbf{x}}_{rand}$. Then, $\hat{\mathbf{x}}_{rand}$ is passed through the decoder to generate a pseudo-example. To create a balanced training set, for each class that mPFC has learned, we generate $\lceil m \rceil$ pseudo-examples, where $m$ is the average number of examples per class stored in HC. The pseudo-examples are mixed with the data in HC, and the mixture is used to fine-tune mPFC using backpropagation. After consolidation, all units in HC are deleted.

## 4.3 NETWORK SELECTION USING BLA

During prediction, FearNet uses the BLA network (Fig. 3(b)) to determine whether to classify an input $\mathbf{x}$ using HC or mPFC. This can be challenging because if HC has only been trained on one class, it will put all of its probability mass on that class, whereas mPFC will likely be less confident. The output of BLA is given by $A(\mathbf{x})$ and will be a value between 0 and 1, with a 1 indicating mPFC should be used. BLA is trained after each study session using only the data in HC and with pseudo-examples generated with mPFC, using the same procedure described in Sec. 4.2. Instead of using

solely BLA to determine which network to use, we found that combining its output with those of mPFC and HC improved results. The predicted class $\hat{y}$ is computed as

$$\hat{y} = \begin{cases} \arg\max_{k'} P_{HC}\left(C = k'|\mathbf{x}\right) & \text{if } \psi > \max_k P_{mPFC}\left(C = k|\mathbf{x}\right) \\ \arg\max_{k'} P_{mPFC}\left(C = k'|\mathbf{x}\right) & \text{otherwise} \end{cases} \qquad (5)$$

where

$$\psi = \left(1 - A\left(\mathbf{x}\right)\right)^{-1} \max_k P_{HC}\left(C = k|\mathbf{x}\right) A\left(\mathbf{x}\right)$$

$\psi$ is the probability of the class according to HC weighted by the confidence that the associated memory is actually stored in HC. BLA has the same number of layers/units as the mPFC encoder, and uses a logistic output unit. We discuss alternative BLA models in supplemental material.

## 5 EXPERIMENTAL SETUP

**Evaluating Incremental Learning Performance.** To evaluate how well the incrementally trained models perform compared to an offline model, we use the three metrics proposed in Kemker et al. (2018). After each study session $t$ in which a model learned a new class $k$, we compute the model's test accuracy on the new class ($\alpha_{new,t}$), the accuracy on the base-knowledge ($\alpha_{base,t}$), and the accuracy of all of the test data seen to this point ($\alpha_{all,t}$). After all $T$ study sessions are complete, a model's ability to retain the base-knowledge is given by $\Omega_{base} = \frac{1}{T-1} \sum_{t=2}^{T} \frac{\alpha_{base,t}}{\alpha_{offline}}$, where $\alpha_{offline}$ is the accuracy of a multi-layer perceptron (MLP) trained offline (i.e., it is given all of the training data at once). The model's ability to immediately recall new information is measured by $\Omega_{new} = \frac{1}{T-1} \sum_{t=2}^{T} \alpha_{new,t}$. Finally, we measure how well the model does on all available test data with $\Omega_{all} = \frac{1}{T-1} \sum_{t=2}^{T} \frac{\alpha_{all,t}}{\alpha_{offline}}$. The $\Omega_{all}$ metric shows how well new memories are integrated into the model over time. For all of the metrics, higher values indicate superior performance. Both $\Omega_{base}$ and $\Omega_{all}$ are relative to an offline MLP model, so a value of 1 indicates that a model has similar performance to the offline baseline. This allows results across datasets to be better compared. Note that $\Omega_{base} > 1$ and $\Omega_{all} > 1$ only if the incremental learning algorithm is more accurate than the offline model, which can occur due to better regularization strategies employed by different models.

**Datasets.** We evaluate all of the models on three benchmark datasets (Table 1): CIFAR-100, CUB-200, and AudioSet. CIFAR-100 is a popular image classification dataset containing 100 mutually-exclusive object categories, and it was used in Rebuffi et al. (2017) to evaluate iCaRL. All images are $32 \times 32$ pixels. CUB-200 is a fine-grained image classification dataset containing high resolution images of 200 different bird

| | CIFAR-100 | CUB-200 | AudioSet |
|---|---|---|---|
| **Classification Task** | RGB Image | RGB Image | Audio |
| **Classes** | 100 | 200 | 100 |
| **Feature Shape** | 2,048 | 2,048 | 1,280 |
| **Train Samples** | 50,000 | 5,994 | 28,779 |
| **Test Samples** | 10,000 | 5,794 | 5,523 |
| **Train Samples/Class** | 500 | 29-30 | 250-300 |
| **Test Samples/Class** | 100 | 11-30 | 43-62 |

Table 1: Dataset Specifications

species (Welinder et al., 2010). We use the 2011 version of the dataset. AudioSet is an audio classification dataset (Gemmeke et al., 2017). We use the variant of AudioSet used by Kemker et al. (2018), which contains a 100 class subset such that none of the classes were super- or sub-classes of one another. Also, since the AudioSet data samples can have more than one class, the chosen samples had only one of the 100 classes chosen in this subset.

For CIFAR-100 and CUB-200, we extract ResNet-50 image embeddings as the input to each of the models, where ResNet-50 was pre-trained on ImageNet (He et al., 2016). We use the output after the mean pooling layer and normalize the features to unit length. For AudioSet, we use the audio CNN embeddings produced by pre-training the model on the YouTube-8M dataset (Abu-El-Haija et al., 2016). We use the pre-extracted AudioSet feature embeddings, which represent ten second sound clips (i.e., ten 128-dimensional vectors concatenated in order).

**Comparison Models**. We compare FearNet to FEL, GeppNet, GeppNet+STM, iCaRL, and an one-nearest neighbor (1-NN). FEL, GeppNet, and GeppNet+STM were chosen due to their previously reported efficacy at incremental class learning in Kemker et al. (2018). iCARL is explicitly designed for incremental class learning, and represents the state-of-the-art on this problem. We compare against 1-NN due to its similarity to our HC model. 1-NN does not forget any previously observed examples, but it tends to have worse generalization error than parametric methods and requires storing all of the training data.

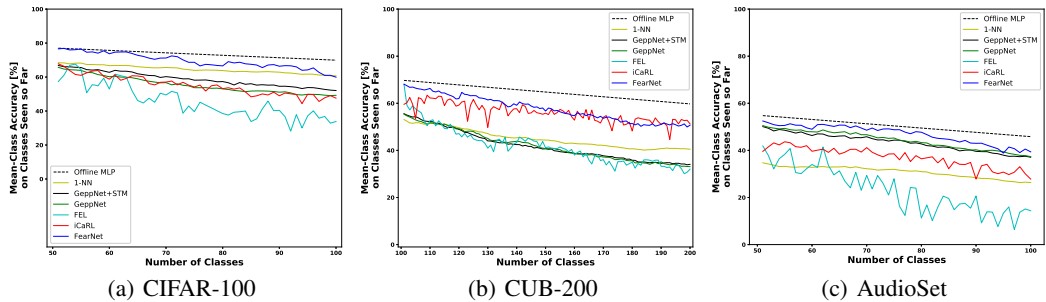

(a) CIFAR-100    (b) CUB-200    (c) AudioSet

Figure 4: Mean-class test accuracy of all classes seen so far.

In each of our experiments, all models take the same feature embedding as input for a given dataset. This required modifying iCaRL by turning its CNN into a fully connected network. We performed a hyperparameter search for each model/dataset combination to tune the number of units and layers (see Supplemental Materials).

**Training Parameters**. FearNet was implemented in Tensorflow. For mPFC and BLA, each fully connected layer uses an exponential linear unit activation function (Clevert et al., 2016). The output of the encoder also connects to a softmax output layer. Xavier initialization is used to initialize all weight layers (Glorot & Bengio, 2010), and all of the biases are initialized to one. BLA's architecture is identical to mPFC's encoder, except it has a logistic output unit, instead of a softmax layer.

mPFC and BLA were trained using NAdam. We train mPFC on the base-knowledge set for 1,000 epochs, consolidate HC over to mPFC for 60 epochs, and train BLA for 20 epochs. Because mPFC's decoder is vital to preserving memories, its learning rate is $1/100$ times lower than the encoder. We performed a hyperparameter search for each dataset and model, varying the model shape (64-1,024 units), depth (2-4 layers), and how often to sleep (see Sec. 6.2). Across datasets, mPFC and BLA performed best with two hidden layers, but the number of units per layer varied across datasets. The specific values used for each dataset are given in supplemental material. In preliminary experiments, we found no benefit to adding weight decay to mPFC, likely because the reconstruction task helps regularize the model.

# 6 EXPERIMENTAL RESULTS

Unless otherwise noted, each class is only seen in one unique study-session and the first base-knowledge study session contains half the classes in the dataset. We perform additional experiments to study how changing the number of base-knowledge classes affects performance in Sec. 6.2. Unless otherwise noted, FearNet sleeps every 10 study sessions across datasets.

## 6.1 STATE-OF-THE-ART COMPARISON

Table 2 shows incremental class learning summary results for all six methods. FearNet achieves the best $\Omega_{base}$ and $\Omega_{all}$ on all three datasets. Fig. 4 shows that FearNet more closely resembles the offline MLP baseline than other methods. $\Omega_{new}$ measures test accuracy on the most recently trained class. [1] For FearNet, this measures the performance of HC and BLA. $\Omega_{new}$ does not account for how well the class was consolidated into mPFC which happens later during a sleep phase; however, $\Omega_{all}$ does account for this. FEL achieves a high $\Omega_{new}$ score because it is able to achieve nearly perfect test accuracy on every new class it learns, but this results in forgetting more quickly than FearNet. 1-NN is similar to our HC model; but on its own, it fails to generalize as well as FearNet, is memory inefficient, and is slow to make predictions. The final mean-class test accuracy for the offline MLP used to normalize the metrics is 69.9% for CIFAR-100, 59.8% for CUB-200, and 45.8% for AudioSet.

---

[1] $\Omega_{new}$ is not scaled with $\alpha_{offline}$, so it does not have the same scale as $\Omega_{base}$ and $\Omega_{all}$.

| Model | CIFAR-100 | | | CUB-200 | | | AudioSet | | | Mean | |
|---|---|---|---|---|---|---|---|---|---|---|---|
| | $\Omega_{base}$ | $\Omega_{new}$ | $\Omega_{all}$ | $\Omega_{base}$ | $\Omega_{new}$ | $\Omega_{all}$ | $\Omega_{base}$ | $\Omega_{new}$ | $\Omega_{all}$ | $\Omega_{base}$ | $\Omega_{all}$ |
| **1-Nearest Neighbor** | 0.878 | 0.648 | 0.879 | 0.746 | 0.434 | 0.694 | 0.655 | 0.269 | 0.613 | 0.760 | 0.729 |
| **GeppNet+STM** | 0.866 | 0.408 | 0.800 | 0.764 | 0.204 | 0.645 | 0.941 | 0.372 | 0.861 | 0.857 | 0.769 |
| **GeppNet** | 0.833 | 0.529 | 0.754 | 0.727 | 0.558 | 0.645 | 0.932 | 0.499 | 0.879 | 0.831 | 0.759 |
| **FEL** | 0.707 | 0.999 | 0.619 | 0.702 | 0.976 | 0.641 | 0.491 | 1.000 | 0.456 | 0.633 | 0.572 |
| **iCaRL** | 0.746 | 0.807 | 0.749 | 0.942 | 0.547 | 0.864 | 0.740 | 0.487 | 0.733 | 0.801 | 0.782 |
| **FearNet** | 0.927 | 0.824 | **0.947** | 0.924 | 0.598 | **0.891** | 0.962 | 0.455 | **0.932** | 0.938 | **0.923** |

Table 2: State-of-the-art comparison on CIFAR-100, CUB-200, and AudioSet. The best $\Omega_{all}$ for each dataset are in **bold**. $\Omega_{base}$ and $\Omega_{all}$ are normalized by the offline MLP baseline.

| | CIFAR-100 | | CUB-200 | | AudioSet | |
|---|---|---|---|---|---|---|
| | Oracle | With BLA | Oracle | With BLA | Oracle | With BLA |
| $\Omega_{base}$ | 0.965 | 0.927 | 0.968 | 0.924 | 0.970 | 0.962 |
| $\Omega_{new}$ | 0.912 | 0.824 | 0.729 | 0.598 | 0.701 | 0.455 |
| $\Omega_{all}$ | 1.002 | 0.947 | 0.936 | 0.891 | 0.972 | 0.932 |

Table 3: FearNet performance when the location of the associated memory is known using an oracle versus using BLA.

## 6.2 Additional Experiments

**Novelty Detection with BLA**. We evaluated the performance of BLA by comparing it to an oracle version of FearNet, i.e., a version that knew if the relevant memory was stored in either mPFC or HC. Table 3 shows that FearNet's BLA does a good job at predicting which network to use; however, the decrease in $\Omega_{new}$ suggests BLA is sometimes using mPFC when it should be using HC.

**When should the model sleep?** To study how the frequency of memory consolidation affects FearNet's performance, we trained FearNet on CUB-200 and varied the sleep frequency from 1-15 study sessions. When FearNet increases the number of classes it learns before sleeping (Fig. 5), it is better able to retain its base-knowledge, but this reduces its ability to recall new information. In humans, sleep deprivation is known to impair new learning (Yoo et al., 2007), and that forgetting occurs during sleep (Poe, 2017). Each time FearNet sleeps, the mPFC weights are perturbed which can cause it to gradually forget older memories. Sleeping less causes HC's recall performance to deteriorate.

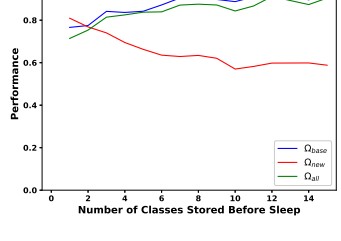

Figure 5: FearNet performance as the sleep frequency decreases.

**Multi-Modal Incremental Learning**. As shown in Sec. 6.1, FearNet can incrementally learn and retain information from a single dataset, but how does it perform if new inputs differ greatly from previously learned ones? This scenario is one of the first shown to cause catastrophic forgetting in MLPs. To study this, we trained FearNet to incrementally learn CIFAR-100 and AudioSet, which after training is a 200-way classification problem. To do this, AudioSet's features are zero-padded to make them the same length as CIFAR-100s. Table 4 shows the performance of FearNet for three separate training paradigms: 1) FearNet learns CIFAR-100 as the base-knowledge and then incrementally learns AudioSet; 2)

| | Base-Knowledge | | |
|---|---|---|---|
| | CIFAR-100 | AudioSet | 50/50 Mix |
| $\Omega_{base}$ | 0.995 | 0.845 | 0.837 |
| $\Omega_{new}$ | 0.693 | 0.903 | 0.822 |
| $\Omega_{all}$ | 0.854 | 0.634 | 0.820 |

Table 4: Multi-modal incremental learning experiment. FearNet was trained with various base-knowledge sets (column-header) and then incrementally trained on all remaining data.

FearNet learns AudioSet as the base-knowledge and then incrementally learns CIFAR-100; and 3) the base-knowledge contains a 50/50 split from both datasets with FearNet incrementally learning the remaining classes. Our results suggest FearNet is capable of incrementally learning multi-modal information, if the model has a good starting point (high base-knowledge); however, if the model starts with lower base-knowledge

performance (e.g., AudioSet), the model struggles to learn new information incrementally (see Supplemental Material for detailed plots).

**Base-Knowledge Effect on Performance**. In this section, we examine how the size of the base-knowledge (i.e., number of classes) affects FearNet's performance on CUB-200. To do this, we varied the size of the base-knowledge from 10-150 classes, with the remaining classes learned incrementally. Detailed plots are provided in the Supplemental Material. As the base-knowledge size increases, there is a noticeable increase in overall model performance because 1) mPFC has a better learned representation from a larger quantity of data and 2) there are not as many incremental learning steps remaining for the dataset, so the base-knowledge performance is less perturbed.

## 7   DISCUSSION

FearNet's mPFC is trained to both discriminate examples and also generate new examples. While the main use of mPFC's generative abilities is to enable psuedorehearsal, this ability may also help make the model more robust to catastrophic forgetting. Gillies (1991) observed that unsupervised networks are more robust (but not immune) to catastrophic forgetting because there are no target outputs to be forgotten. Since the pseudoexample generator is learned as a unsupervised reconstruction task, this could explain why FearNet is slow to forget old information.

Table 5 shows the memory requirements for each model in Sec. 6.1 for learning CIFAR-100 and a hypothetical extrapolation for learning 1,000 classes. This chart accounts for a fixed model capacity and storage of any data or class statistics. FearNet's memory footprint is comparatively small because it only stores class statistics rather than some or all of the raw training data, which makes it better suited for deployment.

| Model | 100 Classes | 1,000 Classes |
|---|---|---|
| **1-NN** | 4.1 GB | 40.9 GB |
| **GeppNet+STM** | 4.1 GB | 41.0 GB |
| **GeppNet** | 4.1 GB | 41.0 GB |
| **FEL** | 272.5 MB | 395.0 MB |
| **iCaRL** | 17.6 MB | 166.0 MB |
| **FearNet** | 10.7 MB | 74.4 MB |

Table 5: Memory requirements to train CIFAR-100 and the amount of memory that would be required if these models were trained up to 1,000 classes.

An open question is how to deal with storage and updating of class statistics if classes are seen in more than one study session. One possibility is to use a running update for the class means and covariances, but it may be better to favor the data from the most recent study session due to learning in the autoencoder.

FearNet assumed that the output of the mPFC encoder was normally distributed for each class, which may not be the case. It would be interesting to consider modeling the classes with a more complex model, e.g., a Gaussian Mixture Model. Robins (1995) showed that pseudorehearsal worked reasonably well with randomly generated vectors because they were associated with the weights of a given class. Replaying these vectors strengthened their corresponding weights, which could be what is happening with the pseudo-examples generated by FearNet's decoder.

The largest impact on model size is the stored covariance matrix $\Sigma_c$ for each class. We tested a variant of FearNet that used a diagonal $\Sigma_c$ instead of a full covariance matrix. Table 6 shows that performance degrades, but FearNet still works.

| | Full Covariance | Diagonal Covariance |
|---|---|---|
| $\Omega_{base}$ | 0.942 | 0.781 |
| $\Omega_{new}$ | 0.805 | 0.877 |
| $\Omega_{all}$ | 0.959 | 0.800 |
| **Model Size** | 10.7 MB | 3.8 MB |

Table 6: Using a diagonal covariance matrix for FearNet's class statistics instead of a full covariance matrix on CIFAR-100.

FearNet can be adapted to other paradigms, such as unsupervised learning and regression. For unsupervised learning, FearNet's mPFC already does a form of it implicitly. For regression, this would require changing mPFC's loss function and may require grouping input feature vectors into similar collections. FearNet could also be adapted to perform the supervised data permutation experiment performed by Goodfellow et al. (2013) and Kirkpatrick et al. (2017). This would likely require storing statistics from previous permutations and classes. FearNet would sleep between learning different permutations; however, if the number of classes was high, recent recall may suffer.

## 8 CONCLUSION

In this paper, we proposed a brain-inspired framework capable of incrementally learning data with different modalities and object classes. FearNet outperforms existing methods for incremental class learning on large image and audio classification benchmarks, demonstrating that FearNet is capable of recalling and consolidating recently learned information while also retaining old information. In addition, we showed that FearNet is more memory efficient, making it ideal for platforms where size, weight, and power requirements are limited. Future work will include 1) integrating BLA directly into the model (versus training it independently); 2) replacing HC with a semi-parametric model; 3) learning the feature embedding from raw inputs; and 4) replacing the pseduorehearsal mechanism with a generative model that does not require the storage of class statistics, which would be more memory efficient.

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

# A  SUPPLEMENTAL MATERIAL

## A.1  MODEL HYPERPARAMETERS

Table S1 shows the training parameters for the FearNet model for each dataset. We also experimented with various dropout rates, weight decay, and various activation functions; however, weight decay did not work well with FearNet's mPFC.

| Hyperparameter | Values |
|---:|:---:|
| Learning Rate | $2 \cdot 10^{-3}$ |
| Mini-Batch Size | 450 (AudioSet & CIFAR-100) 
 200 (CUB-200) |
| mPFC Base-Knowledge Epochs | 1,000 |
| Memory Consolidation Epochs | 60 |
| BLA Training Epochs | 20 |
| Hidden Layer Size | CIFAR-100: $[140, 130]$ 
 CUB-200: $[350, 300]$ 
 AudioSet: $[300, 100]$ |
| Sleep Frequency | 10 (see Sec. 6.2) |
| Dropout Rate | 0.25 |
| Unsupervised Loss Weights ($\lambda$) | $\left[10^4, 1.0, 0.1\right]$ |
| Hidden Layer Activation | Exponential Linear Units |
| Weight Decay | 0.0 |

Table S1: FearNet Training Parameters

Table S2 shows the training parameters for the iCaRL framework used in this paper. We adapted the code from the author's GitHub page for our own experiments. The ResNet-18 convolutional neural network was replaced with a fully-connected neural network. We experimented with various regularization strategies to increase the initial base-knowledge accuracy with weight decay working

the best. The values that are given as a range of values are the hyperparameter search spaces.

| Hyperparameter | Values |
|---|---|
| Learning Rate | $2 \cdot 10^{-3}$ |
| Mini-Batch Size | 450 |
| Exemplars per Class (EPC) | 20 |
| Hidden Layer Size | 64-1024 |
| Number of Hidden Layers | 2-4 |
| Dropout Rate | $[0.5, 0.75, 1.00]$ |
| Hidden Layer Activation | ReLU |
| Weight Decay | $\left[0.0, 10^{-5}, 10^{-4}, 5 \cdot 10^{-4}\right]$ |

Table S2: iCaRL Training Parameters

Table S3 shows the training parameters for GeppNet and GeppNet+STM. Parameters not listed here are the default parameters defined by Gepperth & Karaoguz (2016). The values that are given as a range of values are the hyperparameter search spaces.

| Hyperparameter | Values |
|---|---|
| SOM Lattice Shape ($N$) | 20-36 |
| Non-Linearity Suppression Threshold ($\theta$) | 0.1-0.75 |
| Incremental Class Learning Iterations ($T_{inc2} - T_{inc1}$) | $[2,000, 20,000]$ |

Table S3: GeppNet Training Parameters

Table S4 shows the training parameters for the Fixed Expansion Layer (FEL). The number of units in the FEL layer is given by

$$\text{FEL Units} = \frac{H^2 + HK}{K} \tag{6}$$

where $H$ is the number of units in the first hidden-layer and $K$ is the maximum number of classes in the dataset. The values that are given as a range of values are the hyperparameter search spaces.

| Hyperparameter | Values |
|---|---|
| Hidden Layer Size ($H$) | 64-1800 |
| FEL Layer Size | See Equation 6 |
| Number of Hidden Layers | 2 |
| Mini-Batch Size | 8 |
| Initial Learning Rate | $10^{-2}$ |

Table S4: FEL Training Parameters

## A.2 ICARL PERFORMANCE WITH MORE EXEMPLARS

Table S5 provides additional experimental results for when there are more exemplars per class (EPC) for the iCaRL framework. Rebuffi et al. (2017) used 20 EPC in their original paper; however, we increased the number to 100 EPC to see if storing more training data helped iCaRL. Although a higher EPC does increase iCaRL performance, it still does not outperform FearNet. Note that CUB-200 only has about 30 training samples per class, so iCaRL is storing the entire training set for 100 EPC. Our main results use the default value of 20.

## A.3 BLA VARIANTS

Our BLA model is a classifier that determines whether a prediction should be made using HC (recent memory) or mPFC (remote memory). An alternative approach would be to use an outlier detection algorithm that determines whether the data being processed by a sub-network is an outlier for that sub-network and should therefore be processed by the other sub-network. To explore this alternative BLA formulation, we experimented with three outlier detection algorithms: 1) one-class support vector machine (SVM) (Schölkopf et al., 2001), 2) determining if the data fits into a Gaussian distribution using a minimum covariance determinant estimation (i.e., elliptical envelope) (Rousseeuw

| Model | CIFAR-100 | | | CUB-200 | | | AudioSet | | | Mean | |
|---|---|---|---|---|---|---|---|---|---|---|---|
| | $\Omega_{base}$ | $\Omega_{new}$ | $\Omega_{all}$ | $\Omega_{base}$ | $\Omega_{new}$ | $\Omega_{all}$ | $\Omega_{base}$ | $\Omega_{new}$ | $\Omega_{all}$ | $\Omega_{base}$ | $\Omega_{all}$ |
| iCaRL (20 EPC) | 0.746 | 0.807 | 0.749 | 0.942 | 0.547 | 0.864 | 0.740 | 0.487 | 0.733 | 0.801 | 0.782 |
| iCaRL (100 EPC) | 0.842 | 0.719 | 0.822 | 0.951 | 0.554 | 0.882 | 0.820 | 0.419 | 0.771 | 0.871 | 0.825 |
| FearNet | 0.927 | 0.824 | **0.947** | 0.924 | 0.598 | **0.891** | 0.962 | 0.455 | **0.932** | **0.938** | **0.923** |

Table S5: iCaRL's performance when the stored EPC is increased from 20 to 100.

& Driessen, 1999), and 3) the isolation forest (Liu et al., 2008). All three of these methods set a rejection criterion for if the test sample exists in HC; whereas the binary MLP reports a probability on how likely the test sample resides in HC. Table S6 compares these individual methods. Isolation Forest and Elliptic Envelope seem to prefer the data in HC, one-class SVM prefers the data in mPFC, and our binary MLP worked best at choosing the correct sub-network to use.

| BLA Method | $\Omega_{base}$ | $\Omega_{new}$ | $\Omega_{all}$ |
|---|---|---|---|
| Isolation Forest | 0.328 | 0.823 | 0.368 |
| Elliptic Envelope | 0.518 | 0.823 | 0.541 |
| One-Class SVM | 0.718 | 0.433 | 0.702 |
| Binary MLP | 0.927 | 0.924 | 0.947 |

Table S6: Performance of different BLA variants.

## A.4 FEARNET ALGORITHM

Pseudocode for FearNet's training and prediction algorithms are given in Algorithms 1 and 2 respectively. The variables match the ones defined in the paper.

---

**Algorithm 1:** FearNet Training

**Data:** X,y
**Classes/Study-Sessions**: T;
**K**: Sleep Frequency;
Initialize mPFC with base-knowledge;
Store $\mu_t, \Sigma_t$ for each class in the base-knowledge;
**for** $c \leftarrow T/2$ **to** T **do**
  Store $X, y$ for class $c$ in HC;
  **if** $c \% K == 0$ **then**
    Fine-tune mPFC with $X, y$ in HC and pseudo-examples generated by mPFC decoder;
    Update $\mu_t, \Sigma_t$ for all classes seen so far;
    Clear HC;
  **else**
    Update BLA;

---

**Algorithm 2:** FearNet Prediction

**Data:** X
$A(\mathbf{X}) \leftarrow P_{BLA}(C = 1|\mathbf{X})$;
$\psi \leftarrow \frac{\max_k P_{HC}(C=k|\mathbf{X})A(\mathbf{X})}{1-A(\mathbf{X})}$;
**if** $\psi > \max_k P_{mPFC}(C = k|\mathbf{X})$ **then**
  return
    $\arg \max_{k'} P_{HC}(C = k'|\mathbf{X})$;
**else**
  return
    $\arg \max_{k'} P_{mPFC}(C = k'|\mathbf{X})$;

---

## A.5 MULTI-MODAL LEARNING EXPERIMENT

Fig. S1 shows the plots for the multi-modal experiments in Sec. 6.2. The three base-knowledge experiments were 1) CIFAR-100 is the base-knowledge and AudioSet is trained incrementally, 2) AudioSet is the base-knowledge and then AudioSet is trained incrementally, and 3) the base-knowledge is a 50/50 mix of the two datasets and then the remaining classes are trained incrementally. For all three base-knowledge experiments, we show the mean-class accuracy on the base-knowledge and the entire test set. FearNet works well when it adequately learns the base-knowledge (Experiment #1 and #3); however, when FearNet learns it poorly, incremental learning deteriorates.

## A.6 BASE-KNOWLEDGE EFFECT ON PERFORMANCE

Fig. S2 shows the effect of the base-knowledge's size on FearNet's performance. As expected, $\Omega_{base}$ increases because there are not as many sleep phases to overwrite existing base-knowledge. $\Omega_{new}$

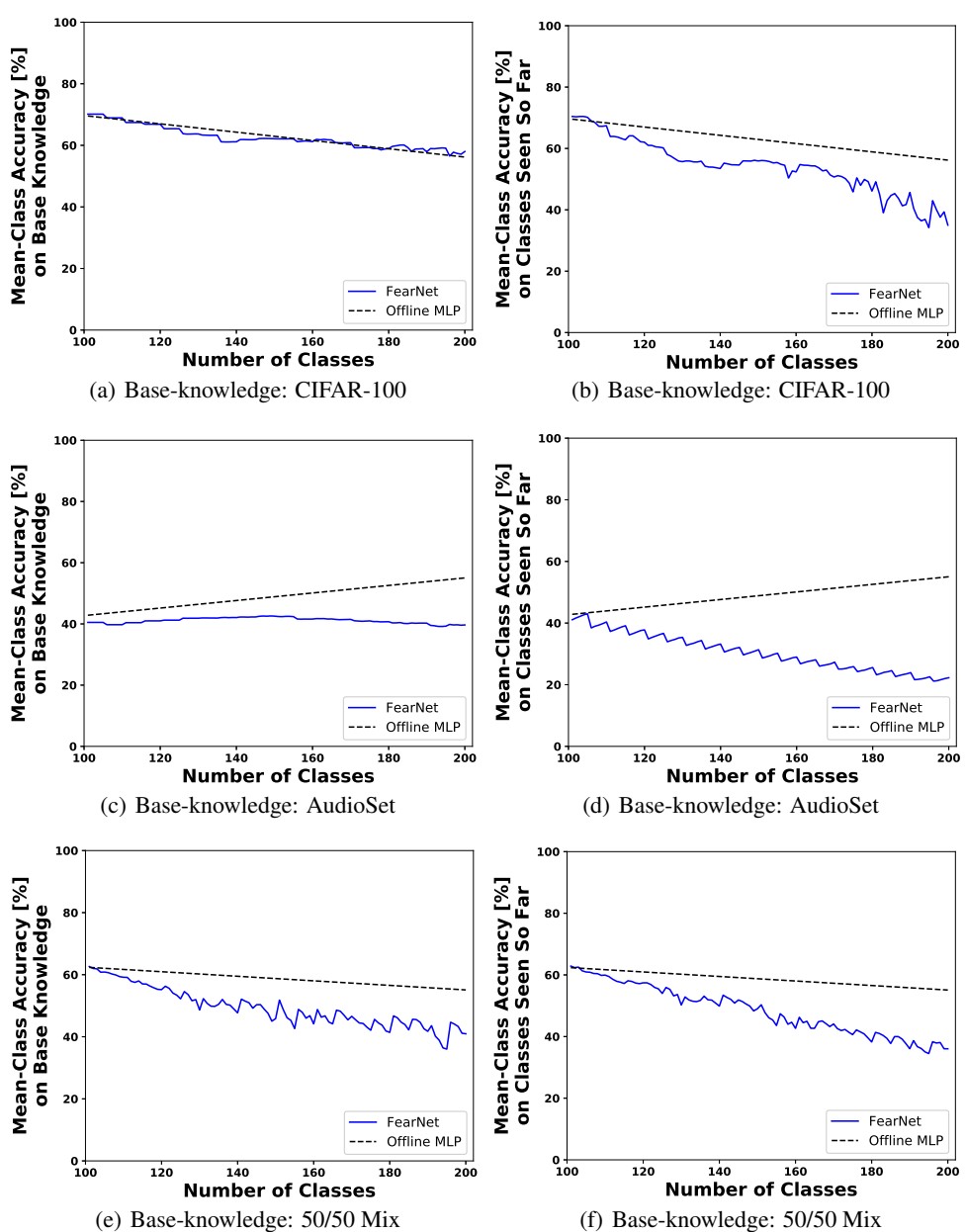

Figure S1: Detailed plots for the multi-modal experiment. The top row is when the base-knowledge was CIFAR-100, the middle row is when the base-knowledge was AudioSet, and the bottom row is when the base-knowledge was a 50/50 mix from the two datasets. The left column represents the mean-class accuracy on the base-knowledge test set and the right column computes mean-class accuracy on the entire test set.

remains relatively even because the size of the base-knowledge has no effect on the HC model's ability to immediately recall new information; however, there is a very slight decrease that corresponds to the BLA model erroneously favoring mPFC in a few cases. Most importantly, $\Omega_{all}$ sees an increase in performance because; like $\Omega_{base}$, there are not as many sleep phases to perturb older memories in mPFC.

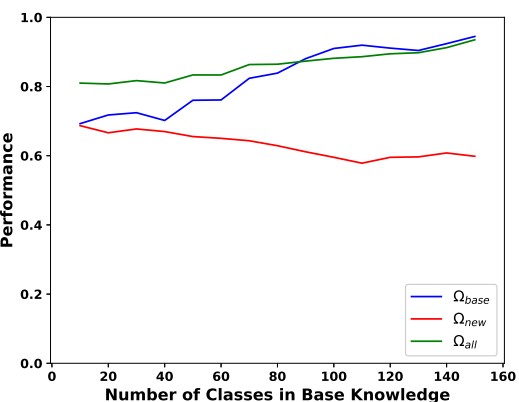

Figure S2: FearNet performance as a function of base-knowledge size.

