# OpenReview forum: "FearNet: Brain-Inspired Model for Incremental Learning"
_ICLR.cc/2018/Conference — Accept (Poster)_

### Official Review · AnonReviewer3 · 2017-11-26
**The paper presents an interesting problem of incremental classification inspired by the dual memory-system of brain. I feel the paper  explicitly describes the problem and explains the proposed methodology in great detail.**

**Rating:** 7
**Confidence:** 4

**Review:**

Quality: The paper presents a novel solution to an incremental classification problem based on a dual memory system. The proposed solution is inspired by the memory storage mechanism in brain.

Clarity: The problem has been clearly described and the proposed solution is described in detail. The results of numerical experiments and the real data analysis are satisfactory and clearly shows the superior performance of the method compared to the existing ones.

Originality: The solution proposed is a novel one based on a dual memory system inspired by the memory storage mechanism in brain. The memory consolidation is inspired by the mechanisms that occur during sleep. The numerical experiments showing the FearNet performance with sleep frequency also validate the comparison with the brain memory system.

Significance: The work discusses a significant problem of incremental classification. Many of the shelf deep neural net methods require storage of previous training samples too and that slows up the application to larger dataset. Further the traditional deep neural net also suffers from the catastrophic forgetting. Hence, the proposed work provides a novel and scalable solution to the existing problem.

pros: (a) a scalable solution to the incremental classification problem using a brain inspired dual memory system
          (b) mitigates the catastrophic forgetting problem using a memory consolidation by pseudorehearsal.
          (c) introduction of a subsystem that allows which memory system to use for the classification

cons: (a)  How FearNet would perform if imbalanced classes are seen in more than one study sessions?
          (b) Storage of class statistics during pseudo rehearsal could be computationally expensive. How to cope with that?
          (c) How FearNet would handle if there are multiple data sources?

---

> ### Author Response · Authors · 2017-12-20
> **Comments for Reviewer #3**
>
>
> Reviewer #3: How FearNet would perform if imbalanced classes are seen in more than one study sessions?
>
> Authors: FearNet generates a balanced number of pseudoexamples during its sleep phase and when updating BLA, so class imbalance is not an issue.  To test this, we did an experiment with CIFAR-100 where we selected a random number of samples from each class (20-500) so that the class distribution was imbalanced.  We would expect a slight degradation in performance because we aren’t using as many samples to train FearNet as we did in the paper (i.e., the model doesn’t generalize as well for the test set.)  The results (\omega_{base} = 0.884, \omega_{new} = 0.729, and \omega_{all} = 0.897) indicate that FearNet is robust to imbalanced class distributions.
>
> Reviewer #3: Storage of class statistics during pseudo rehearsal could be computationally expensive. How to cope with that?
>
> Authors: We agree that storing class statistics is a major bottleneck, but FearNet still manages to be less memory intensive than other models.  In Table 5, we show that the storage cost for FearNet is still lower than previous methods.  In Table 6, we show that FearNet still outperforms other methods when only a diagonal covariance matrix is stored for each class, while decreasing storage costs by 65%.
>
> Reviewer #3: How FearNet would handle if there are multiple data sources?
>
> Authors: We assume that the reviewer is referring to FearNet’s ability to handle multiple data modalities.  In Table 4, we explored FearNet’s ability to simultaneously learn audio and visual information.  The results showed that FearNet was able to simultaneously learn datasets with very different data representations.

---

> > ### Comment · AnonReviewer3 · 2018-01-12
> > **Well revised and the previous issues have been clarified**
> >
> > I am happy with the revision! My concerns regarding the FearNet mechanism have been properly addressed. The issue of class imbalance, computational hurdles in storage and the issue of multiple data modalities have been addressed appropriately.

---

### Official Review · AnonReviewer2 · 2017-11-27
**Interesting cognitive science take, tested on modern datasets**

**Rating:** 7
**Confidence:** 2

**Review:**

I quite liked the revival of the dual memory system ideas and the cognitive (neuro) science inspiration. The paper is overall well written and tackles serious modern datasets, which was impressive, even though it relies on a pre-trained, fixed ResNet (see point below).

My only complaint is that I felt I couldn’t understand why the model worked so well. A better motivation for some of the modelling decisions would be helpful. For instance, how much the existence (and training) of a BLA network really help — which is a central new part of the paper, and wasn’t in my view well motivated. It would be nice to compare with a simpler baseline, such as a HC classifier network with reject option. I also don’t really understand why the proposed pseudorehearsal works so well. Some formal reasoning, even if approximate, would be appreciated.

Some additional comments below:

- Although the paper is in general well written, it falls on the lengthy side and I found it difficult at first to understand the flow of the algorithm. I think it would be helpful to have a high-level pseudocode presentation of the main steps.

- It was somewhat buried in the details that the model actually starts with a fixed, advanced feature pre-processing stage (the ResNet, trained on a distinct dataset, as it should). I’m fine with that, but this should be discussed. Note that there is evidence that the neuronal responses in areas as early as V1 change as monkeys learn to solve discrimination tasks. It should be stressed that the model does not yet model end-to-end learning in the incremental setting.

- p. 4, Eq. 4, is it really necessary to add a loss for the intermediate layers, and not only for the input layer? I think it would be clearer to define the \mathcal{L} explictily somewhere. Also, shouldn’t the sum start at j=0?

---

> ### Author Response · Authors · 2017-12-20
> **Comments for Reviewer #2**
>
>
> Reviewer #2: My only complaint is that I felt I couldn’t understand why the model worked so well. A better motivation for some of the modelling decisions would be helpful. For instance, how much the existence (and training) of a BLA network really help — which is a central new part of the paper, and wasn’t in my view well motivated. It would be nice to compare with a simpler baseline, such as a HC classifier network with reject option.
>
> Authors: We do explore how the BLA effects FearNet performance in an ablation study shown in Table 3.  We actually tried different variants for BLA before settling on the model that we used in the paper. We have included the results of the other variants in the supplemental material to help justify our decisions.
>
> Reviewer #2: I also don’t really understand why the proposed pseudorehearsal works so well. Some formal reasoning, even if approximate, would be appreciated.
>
> Authors: Rehearsal and psuedorehearsal are old ideas from the 1990s. We have added more justification for why they help alleviate catastrophic forgetting in Section 2 and in the discussion.
>
> Reviewer #2: Although the paper is in general well written, it falls on the lengthy side and I found it difficult at first to understand the flow of the algorithm. I think it would be helpful to have a high-level pseudocode presentation of the main steps.
>
> Authors: The high-level pseudocode for FearNet’s train and predict functionality is a great idea.  We have included this in the supplemental material of our revised version.
>
> Reviewer #2: It was somewhat buried in the details that the model actually starts with a fixed, advanced feature pre-processing stage (the ResNet, trained on a distinct dataset, as it should). I’m fine with that, but this should be discussed. Note that there is evidence that the neuronal responses in areas as early as V1 change as monkeys learn to solve discrimination tasks. It should be stressed that the model does not yet model end-to-end learning in the incremental setting.
>
> Authors: We have included the following sentence in the beginning of Section 4, “In this paper, we use pre-trained embeddings of the input (e.g., ResNet).”  We think representation learning is an important next step, and it will be incorporated into FearNet 2.0, which is currently in its early planning stages.
>
> Reviewer #2: p. 4, Eq. 4, is it really necessary to add a loss for the intermediate layers, and not only for the input layer? I think it would be clearer to define the \mathcal{L} explictily somewhere. Also, shouldn’t the sum start at j=0?
>
> Authors: Thank you for pointing that out.  We have fixed Eq. 4 and defined the \mathcal{L} term to make it clear that we are computing the MSE loss between the output of each hidden layer and the input/output of the mPFC autoencoder.  The rationale for using MSE losses at the intermediate layers stem from Valpola (2015), where he showed that errors in deeper layers had a harder time being corrected because they were further away from the training signal (i.e., data layer).  Adding the multi-layer loss forces the autoencoder to correct errors at every layer.  A good autoencoder fit is important for our framework because it is directly related to the fidelity of the pseudoexamples being generated for sleep phases.

---

### Official Review · AnonReviewer1 · 2017-11-30
**Suprizingly good results with an rather simple architecture. Is the comparison to SotA fair??**

**Rating:** 6
**Confidence:** 2

**Review:**


This paper addresses the problem of incremental class learning with brain inspired memory system. This relies on 1/ hippocampus like system relying on a temporary memory storage and probabilistic neural network classifier, 2/ a prefrontal cortex-like ladder network architecture, performing joint autoencoding and classification, 3/ an amygdala-like classifier that combines the decision of both structures. The experiments suggests that the approach performs better than state-of-the-art incremental learning approaches, and approaches offline learning.
The paper is well written. The main issue I have with the approach is the role of the number of examples stored in hippocampus and its implication for the comparison to state-of-the art approaches.
Comments:
It seems surprising to me that the network manages to outperform other approaches using such a simplistic network for hippocampus (essentially a Euclidian distance based classifier). I assume that the great performance is due to the fact that a lot of examples per classes are stored in hippocampus. I could not find an investigation of the effect of this number on the performance. I assume this number corresponds to the mini-batch size (450). I would like that the authors elaborate on how fair is the comparison to methods such as iCaRL, which store very little examples per classes according to Fig. 2. I assume the comparison must take into account the fact that FearNet stores permanently relatively large covariance matrices for each classes.
Overall, the hippocampus structure is the weakness of the approach, as it is so simple that I would assume it cannot adapt well to increasingly complex tasks. Also, making an analogy with hippocampus for such architecture seems a bit exaggerated.

---

> ### Author Response · Authors · 2017-12-20
> **Comments for Reviewer #1**
>
>
> Reviewer #1: It seems surprising to me that the network manages to outperform other approaches using such a simplistic network for hippocampus (essentially a Euclidian distance based classifier). I assume that the great performance is due to the fact that a lot of examples per classes are stored in hippocampus.
>
> Authors: Our comparison to the state-of-the-art is valid, and we have thoroughly checked our results.  The HC network is simple, and we are comparing not only it, but the entire network to the state-of-the-art.  We chose to implement HC using nearest neighbor density estimation because it is able to make inferences from new data immediately without expensive loops through the training data and often works well when data is scarce (low-shot learning), and it has the perfect properties for enabling us to use pseudorehearsal for consolidating information from HC to mPFC. Psuedorehearsal requires mixing the raw recently observed data with generated examples of data observed long ago.  Our HC model can be thought of as a simple buffer that can enable inference to be made until the information is transferred to mPFC, at which point the HC model is erased. After “sleeping,” FearNet does not store old values in HC because those memories now reside in the mPFC network.  FearNet uses its BLA module to determine where the memory resides.  Since FearNet could erroneously predict the network where the memory resides, we don’t believe that HC artificially inflates FearNet performance.  We tested this by performing the incremental learning experiment for the 1-nearest neighbor (1-NN) for all three datasets (see Table 2 in the revised manuscript). FearNet outperformed 1-NN because 1-NN was unable to generalize to the test data as well as FearNet.  Additionally, compared to FearNet, 1-NN is significantly less memory efficient (Table 5) and very slow at making predictions.
>
> Reviewer #1: I could not find an investigation of the effect of this number on the performance. I assume this number corresponds to the mini-batch size (450).
>
> Authors: We did investigate FearNet performance as a function of how many classes are learned (stored in HC) before its sleep phase is performed (see Fig. 5 in the discussion).  The mini-batch is only for 1) the sleep phase and 2) updating BLA.  To make this clearer, we added “We investigate FearNet’s performance as a function of how much data is stored in HC in Section 6.2.” to the end of Section 4.1.
>
> Reviewer #1: I would like that the authors elaborate on how fair is the comparison to methods such as iCaRL, which store very little examples per classes according to Fig. 2. I assume the comparison must take into account the fact that FearNet stores permanently relatively large covariance matrices for each classes.
>
> Authors: For CIFAR-100, iCaRL stores 2,000 exemplars for replay.  At the beginning, they are able to store most/all of the exemplars (there are 500 per class available) since the buffer maxes out at 2,000.  As time moves on, that number decreases as it has to make room for new classes.  By the end, there are 20 exemplars per class.  We have re-written the last paragraph in Section 2 to clarify this point.  In comparison, our model stores the mean/covariance matrix for each class, and then generates new “exemplars” (pseudoexamples) during sleep. Note that our method still outperforms iCaRL and other methods when only a diagonal covariance is stored, as discussed in Section 6.2 (see Table 6).  Using MLP type architectures for iCaRL and FearNet, we showed that storing class statistics is still more memory efficient than storing these exemplars (see Table 5).  Our future work will focus on using generative models that don’t require class statistics for pseudorehearsal.
>
> Reviewer #1: Overall, the hippocampus structure is the weakness of the approach, as it is so simple that I would assume it cannot adapt well to increasingly complex tasks. Also, making an analogy with hippocampus for such architecture seems a bit exaggerated.
>
> Authors: We agree that HC could be improved, and we included in our future work that we want to replace HC with a semi-parametric model, instead of an entirely non-parametric model.  We also agree that the low-level operations that occur in the individual FearNet modules (e.g., HC) are not entirely analogous to operations that occur in the brain; and to be fair, we don’t make that claim.  Our main inspiration for FearNet is 1) the brain’s dual-memory architecture for rapid acquisition of new information and long-term storage of old information, 2) how mammalian brains consolidate recent memories to long term storage during sleep, and 3) the recent and remote recall pathways that BLA uses.

---

### Author Response · Authors · 2017-12-20
**Revised Manuscript Available!**

First, we would like to thank the reviewers for their valuable feedback.  Their comments have helped us to improve the original manuscript.  All three reviewers expressed that they liked our new brain-inspired algorithm for incremental class learning.  We identified two main issues that they raised: 1) the reviewers wanted more justification for architectural decisions (with a focus on HC and BLA); and 2) the reviewers wanted more explanation for why pseudorehearsal works for mitigating catastrophic forgetting during incremental class learning. Additionally, the reviewers suggested a number of minor changes that will make the paper clearer and enable others to better reproduce our work, although we will make all of our code available once the paper is accepted.  We address each reviewer comment individually.  Please let us know if there are any other questions/concerns regarding our revised manuscript.  Thank you!

---

### Decision · Program_Chairs · 2018-01-29
**ICLR 2018 Conference Acceptance Decision**

**Decision:**

Accept (Poster)

**Comment:**

A novel dual memory system inspired by brain for the important incremental learning and very good results.